# Overview and Management of the Most Common Eukaryotic Diseases of Flax (*Linum usitatissimum*)

**DOI:** 10.3390/plants12152811

**Published:** 2023-07-28

**Authors:** Julie Moyse, Sylvain Lecomte, Shirley Marcou, Gaëlle Mongelard, Laurent Gutierrez, Monica Höfte

**Affiliations:** 1Laboratory of Phytopathology, Department of Plants and Crops, Faculty of Bioscience Engineering, Coupure Links 653, 9000 Ghent, Belgium; julie.moyse@ugent.be (J.M.); shirley.marcou@ugent.be (S.M.); 2Centre de Ressources Régionales en Biologie Moléculaire, University of Picardie Jules Verne, UFR Sciences, 33 Rue St-Leu, 80039 Amiens, France; gaelle.mongelard@u-picardie.fr; 3LINEA–Semences, 20 Avenue Saget, 60210 Grandvilliers, France; slecomte@linea-semences.com

**Keywords:** *Linum usitatissimum*, eukaryotic pathogens, disease management, biocontrol, breeding, genetic resistance

## Abstract

Flax is an important crop cultivated for its seeds and fibers. It is widely grown in temperate regions, with an increase in cultivation areas for seed production (linseed) in the past 50 years and for fiber production (fiber flax) in the last decade. Among fiber-producing crops, fiber flax is the most valuable species. Linseed is the highest omega-3 oleaginous crop, and its consumption provides several benefits for animal and human health. However, flax production is impacted by various abiotic and biotic factors that affect yield and quality. Among biotic factors, eukaryotic diseases pose a significant threat to both seed production and fiber quality, which highlights the economic importance of controlling these diseases. This review focuses on the major eukaryotic diseases that affect flax in the field, describing the pathogens, their transmission modes and the associated plant symptoms. Moreover, this article aims to identify the challenges in disease management and provide future perspectives to overcome these biotic stresses in flax cultivation. By emphasizing the key diseases and their management, this review can aid in promoting sustainable and profitable flax production.

## 1. Introduction

Flax (*Linum usitatissimum* L.) is an annual crop from the *Linaceae* family. Flax originates from the Fertile Crescent, where it was first domesticated, but its cultivation has spread out over time. In this species, plant breeding has led to the selection of two agronomic cultivar groups: linseed and fiber flax. Each type has been bred for a different usage. Linseed is a short plant, measuring between 40 and 60 cm in height at harvest, with a high number of ramifications and bolls (fruits). Fiber flax is a taller plant, growing up to 80 to 110 cm in height and displaying a sparsely ramified inflorescence gathered at the top, which contains only a few bolls. Both types are threatened by pests such as fungi, bacteria, viruses, phytoplasmas or insects. However, major flax diseases are caused by eukaryotic microorganisms [1], on which this review will be focused.

Linseed is an oilseed crop producing seeds with a high content of alpha-linolenic acid (ALA), an omega-3 fatty acid. It has been cultivated for more than 10,000 years [2]. Nowadays, linseed is mainly produced in Eastern Europe, Central Asia and Northern America with the Russian Federation, Kazakhstan, Canada and China as the main producing countries [3] (Table 1A). During the last decade, the cultivated areas have increased (from 74,312 ha in 2011 to 129,770 ha in 2021 in Eastern Europe [3]). The highest crop yields are obtained in France, and they have tripled since the early seventies (0.6 ton/ha in 1970 versus 1.9 ton/ha in 2021 [3]). After around 120 days of cultivation in the field, a combine harvester is used to harvest flax bolls. The International Rules for Seed Testing (IRST) recommend flax seeds to be harvested when the humidity level is below 10% to ensure the quality of the extracted oil and prevent the presence of pathogens [4]. Linseed, which is rich in omega-3 fatty acids and lignans—both of which offer health benefits—is used in animal and human feed but also in the production of cosmetics, paint binders and wood protectants.

Fiber flax has been used for its natural bast fiber for more than 30,000 years [5], and its fibers were first twisted to be woven 4000 years ago by the Babylonians [6]. Nowadays, flax fibers are used to produce fabrics, ropes and insulation materials and in polymer formulations. Europe is currently the world leader in fiber flax production, with France being the leading producer [3,7] (Table 1B). The best-quality fibers are produced in Europe, in an area that goes from Caen (Normandy, France) to Amsterdam (the Netherlands), because of the presence of a temperate climate and drained land. Cultivated surfaces of fiber flax are increasing every year in Europe (from 70,000 ha to 162,580 ha in the last decade [3,7]), whereas fiber yield remains a highly variable parameter (e.g., 7.8 tons/ha in 2014 but only 5.3 tons/ha in 2020 [3]) following highly variable weather conditions. After about 100 days of growth in the field, flax is uprooted, and plants remain laid down on soil for several weeks. During this step, called retting, natural bacteria (mostly *Bacillus*, *Clostridium* and *Pseudomonas* genera) and fungi (mostly *Ascomycota* phylum) decompose the cell tissues surrounding fiber bundles, which is crucial for subsequent optimal industrial fiber extraction. Balanced climatic conditions, warm temperature and humidity (heavy night-time dews) are required to complete this step properly. After weeks of retting, the well-dried stalks are harvested in balls and stored until the final step: scutching. This industrial process mechanically separates stem residues (shives) from flax fiber. Every part of the flax plant is usable, which makes flax a waste-free crop. After the scutching step, fiber flax produces 3 to 5% of seeds (mainly used for animal feed, oil production), 15 to 25% of long fibers (spun, woven and can be dyed for diverse uses such as clothing, drapery fabric, technical fabric), 10 to 25% of short fibers or oakum (mainly used in the paper industry and as base material for insulation and polymer products) and 40 to 60% of shives and spangles (mainly used for animal litter, community heating source, chipboard for furniture production, soil cover for vegetation) [8]. Long fibers are the most solid ones; short fibers are broken during the scutching process and are therefore weaker. One hectare of cultivated fiber flax allows one to produce 1200 to 1700 kg of long fibers that can be sold for EUR 3 to EUR 5 per kg in the case of high-quality fibers, which is 10 to 20 times higher than wheat or corn (LINEA Semences, personal communication, 2022).

The production of best-quality fibers and seeds is the result of fine-tuned crop management. The flax-growing cycle is short (from March–April to August in Europe), and the success of the cultivation depends on many different abiotic and biotic factors, among which the most damaging events for flax fiber and seed production are diseases caused by eukaryotic microorganisms [1]. These pathogens can threaten flax cultivation at any step and can be spread by different transmission modes, as shown in Figure 1.

In this review, we present eukaryotic diseases of flax and discuss possible management strategies. The causative pathogens, their lifestyle and affected flax organs are detailed in Table 2. There is no general survey or data on flax yield losses due to the different diseases, but the most damaging effects are due to wilts, rust and pasmo [9]. Managing this broad range of diseases can be challenging because each pathogen has its own characteristics, such as its mode of transmission, the affected plant organ and the developmental stage of the plant when infection occurs. This review will focus on eukaryotic diseases that spread through the air, seeds and soil. Despite there being multiple modes of transmission, each disease is classified by its primary mode of spread.

## 2. Air-Borne Diseases

### 2.1. Powdery Mildew—Golovinomyces orontii/Podosphaera lini/Erysiphe lini/Leveillula taurica

Several powdery mildew species are able to infect flax, including *Erysiphe lini*, *Golovinomyces orontii*, *Leveillula taurica* and *Podosphaera lini* [31]. *Erysiphe lini* and *P. lini* only infect flax. *Erysiphe lini* is known to occur in Japan, while *P. lini* has been reported in Asia and Europe. Also, the wide-host-range pathogen *G. orontii* (often referred to as *Erysiphe cichoracearum* in old records) has been reported in flax from various countries, including Asia, Europe and North America. *Leveillula taurica* is known in a wide range of hosts and has been reported in flax in warm and arid regions [32]. In many reports, the name *Oidium lini* is used for flax powdery mildew. This is an invalid name, and *O. lini* in flax may refer to *E. lini*, *G. orontii* or *P. lini* resulting in many unclear records from different parts of the world. Nowadays, *O. lini* is regarded as a synonym of *P. lini* [33]. All these pathogens are obligate biotrophic ascomycetes and produce similar symptoms: the surface of leaves, stems and sepals are covered with white powdery colonies (Figure 2A). They are composed of white mycelia and conidia but can also include smaller dark sexual fruiting bodies, named chasmothecia, which stay as resting structures until favorable conditions for sporulation are fulfilled. Once spores have been produced, splashing and wind can easily spread the pathogen to other plants around. This disease emerges under hot and dry conditions, particularly toward the end of spring when flax is in the pre-flowering stage [34]. Severe infection causes the senescence of stems and leaves, which ultimately lead to plant death [1].

This disease is commonly controlled by foliar application with systemic QoI (quinone outside inhibitor), SDHI (succinate dehydrogenase inhibitor) and DMI-type (demethylation inhibitor) fungicides [35] (Table 3A). The contact fungicide cyflufenamid is authorized in France and Belgium to control powdery mildew (Table 3A). Over the course of the last decade, powdery mildew resistance has been selected during flax varietal creation, and several European and Canadian cultivars display a good level of resistance against this disease. A major dominant resistance gene to flax powdery mildew, named *Pm1*, has been identified in various Canadian cultivars, while two additional dominant genes have also been postulated [36]. More recently, analysis of the flax genetic resources has allowed researchers to identify several quantitative trait loci (QTLs) related to the resistance to *P. lini* [37]. This knowledge will enable the use of marker-assisted selection in flax breeding, in order to efficiently produce new resistant cultivars [38].

### 2.2. Rust—Melampsora lini

Rust is the most common flax disease in the world, which is caused by the obligate biotrophic basidiomycete *Melampsora lini* [46]. Symptoms of the disease are bright orange powdery pustules and dried scars covering all aerial parts of the plant (leaves, stems and bolls) (Figure 2B,C). Depending on the level of basal genetic flax resistance to rust, a plant defense reaction can occur, leading to dark chlorosis and necrotic halos on the leaves around the orange pustules. *Melampsora lini* is a macrocyclic rust with five possible spore stages. The rust is autoecious: all stages of the life cycle occur on the same host. In the asexual cycle, urediniospores form orange-colored pustules on the leaves that can cause repeated infections. At the end of the growing season, telia are produced, usually on the stem, with thick-walled teliospores that enable the fungus to survive. Teliospores are resistant to adverse climatic conditions, making eradication of the pathogen in a field very difficult. Therefore, the most efficient way to reduce the development and the spread of this disease is the use of flax cultivars owning genetic resistance. The flax—*M. lini* interaction has served as a model for gene-for-gene resistance and has been extensively studied [47]. This resistance was found in the 1990s and subsequently used in breeding programs, leading to the production of resistant cultivars. More than 30 dominant resistance genes have been identified in flax, and many of these genes have been cloned and sequenced [46,48]. Their use has contributed to efficiently eradicating rust from European fields. However, the genetic mechanisms underlying flax resistance against rust remain poorly understood, and studies are still ongoing to improve our knowledge on this disease [49].

## 3. Seed-Borne Diseases

### 3.1. Alternaria Blight—Alternaria linicola

Alternaria blight occurs mostly in Asian linseed fields and can cause up to 60% of yield loss [14]. At the early stage of flax plant development, Alternaria blight causes dark red lesions in seedlings and, in the case of severe infection, leads to the death of plantlets. When they stay alive, infected seedlings are shriveled, and their stems show dark brown spots, as shown in Figure 3A. Dead plant fragments left on the soil are a rich source of primary inoculum, since the pathogen can survive as thick-walled chlamydospores on flax debris and in the soil [50]. The infection can also occur at a later developmental stage of the plant and results in chlorotic spots on leaves which lead to leaf senescence. To avoid the propagation of the disease, the use of pathogen-free seed batches is crucial: the International Seed Testing Association (ISTA) [4] recommends that commercial batches of flax seeds contain less than 1% of seeds contaminated with *A. linicola*. In addition, the presence of plant debris must be avoided in seed batches, since it constitutes the most obvious source of pathogen residues. Cleaning techniques using hot air or steam can be used on flax seeds. But the fungus can also be found inside seeds which makes it impossible to eradicate with such seed cleaning protocols. Seed treatment with fludioxonil is allowed in Belgium and the Netherlands to control Alternaria blight. Overall, the use of resistant cultivars remains the best way to control this disease since the surviving structures in soil are difficult to eradicate and can stay there for years. Recently, QTLs related to Alternaria blight resistance have been identified in flax [51] and are likely to be used for plant breeding in the coming years.

### 3.2. Anthracnose—Colletotrichum lini

*Colletotrichum lini (syn: Colletotrichum linicola)* can cause seedling blight before or after emergence and is responsible for the loss of large cultivated areas [1]. When the flax plant survives the infection, circular dark green to brown lesions appear on cotyledons and first leaves, which finally become senescent (Figure 3B). *Colletotrichum lini* development is favored by warm weather and rain, promoting its spread within a field by splash from one plant to another. Sowing seeds from pathogen-free batches is the best solution to decrease the risk of the emergence of this disease. The IRST protocol [4] advises *C. lini* to be detected in less than 5% of seeds. If not, seed cleaning procedures or fungicide-based coatings are recommended since mycelia of *C. lini* can also be found in the seed coat. Several triazole-based fungicides are authorized outside of Europe to control this disease on flax, but since their foliar application cannot decrease the primary inoculum, this does not represent a long-term solution.

### 3.3. Basal Stem Blight/Foot Rot—Boeremia linicola

*Boeremia linicola* (previously known as *Phoma exigua* f. sp. *linicola*) causes the yellowing of seedlings which can lead to the death of plantlets. This fungus has been detected in almost every flax-cultivating area during the last century [1], and its presence in a field can completely destroy the crop. When the plant survives, its roots exhibit a brown coloration with discolored zones extending toward the upper part. Meanwhile, the aerial part of the plant turns yellow and wilted and displays elongated brown lesions on the stem (Figure 3C,D). Plant death can occur at any stage of the cultivation before flowering. The saprophytic phase of its developmental cycle starts as mycelia inside the seed coat and evolves to pycnidia, which are produced in large numbers in infected or on dead plant tissue. The severity of the disease explains why this microorganism is targeted by the French seed testing protocol [52,53]. In a batch of seeds, *B. exigua* may not be detected in more than 1% of flax seeds (5% for linseed) to be certified. The International Seed Federation (ISF, Switzerland [54]) keeps up to date a list of pests that need to be avoided on trade seeds, in which *B. linicola* is cited as a “regulated pest” on okra (*Abelmoschus esculentus*) and bean (*Phaseolus vulgaris*). However, no seed treatment has shown efficacy against the pathogen, and only a few and barely efficient fungicides are usable in France to face it in flax fields: foliar application of difenoconazole- and boscalid-based formulations (Table 3A).

### 3.4. Browning and Stem Break—Aureobasidium pullulans var. lini

The browning and stem break of flax are caused by a pathogen that used to be called *Polyspora* or *Kabatiella lini* with *Guignardia fulvida* as a teleomorph (syn: *Discosphaerina fulvida* [55]). Based on ITS1 and ITS2 sequences, however, it was shown that *K. lini* should be considered a synonym of *Aureobasidium pullulans* var. *lini* [56]. *A. pullulans* is a ubiquitous black yeast that is commonly found in the phyllosphere of plants and is a potential biocontrol agent of aerial fungal pathogens [57]. In flax, *A. pullulans* causes dark brown circular lesions (Figure 3E) and blight on flax seedlings and stems. Cankers can also occur on stems and even break them (Figure 3F). The pathogen can be found inside the flax seed coat. The emergence and propagation of this seed-borne disease occur when infected seeds are emerging from soil. Infected seedlings can show symptoms of blight or stem break. As flax plants can remain alive despite a broken stem, the detection of this disease remains unclear. It has been reported to be devastating mostly in Europe and Northern America [1,58]. *A. pullulans* can survive on plant debris, including stubble, and reproduce on secondary hosts. To minimize the impact of stem break on flax, sowing in a relatively dry and cool soil is recommended as the development of this fungus is favored by warm and wet environmental conditions. The use of disease-free seed batches helps to decrease the in-field inoculum. Crop rotations with cereals and pulse crops are recommended in Canada [58]. No seed treatment is available and only a few leaf treatments (triazole- or methoxy-acrylate-based) are authorized in France and Belarus with a medium to high risk of resistance (see Table 3A for details).

### 3.5. Gray Mold—Botrytis cinerea

This fungal disease caused by *Botrytis cinerea* induces the damping-off of young seedlings at emergence. In surviving plants, *B. cinerea* produces gray spots (colonies, Figure 3G) and brown to black areas (sclerotia) on the plant stem, which lead to the decay and death of the plant. When the plant stays alive up to the seed production stage, gray colonies also develop on bolls (Figure 3H). Lodging events can increase the severity of the disease and cause major yield losses [1]. Due to its necrotrophic lifestyle (Figure 3I), this fungus can survive for a while on plant debris until finding a new host to infect. That turns out to be quite easy given its wide range of host plants, including more than 200 crop species. In the case of infection in a field, the inoculum pressure increases fast, and the disease becomes difficult to control, causing each year several hundreds of millions of US dollars of crop losses worldwide [59]. For commercial flax seeds in Europe, the IRST protocol [4] advises the detection of *B. cinerea* in less than 5% of seeds. To avoid early infection during the germination stage, flax seeds can be coated with fungicides, using, for example, fludioxonil-based formulations (authorized in Belgium and the Netherlands, Table 3A). When the disease strikes a flax field during the vegetation stage, it is hard to get rid of. Some foliar fungicides are commercialized (tebuconazole formulations, only usable in the Republic of Ireland and Great Britain, or cyprodinil and fluopyram registered in the Netherlands, Table 3A), which have poor efficiency and can only be used at an early developmental stage to avoid any interference with the subsequent retting process.

### 3.6. Pasmo—Septoria linicola (syn.: Mycosphaerella linicola)

Pasmo disease or septoriosis is characterized in flax by brown lesions in seedlings, leaves (Figure 3J), stems and even bolls. Tissue senescence could also occur at a later stage of infection. During seed production, the colonies appear as black spots (pycnidia) developing on bolls (Figure 3K). Pycnidia quickly increase the inoculum by releasing millions of spores under windy or rainy conditions. Pycnidia are also able to survive on organic matter, which constitutes a large reservoir of inoculum. *Septoria linicola*, which causes this disease, has a teleomorph form, named *Mycosphaerella linicola* [21]. The sexual stage plays a significant role in the epidemiology of the disease. Ascospores released from pseudothecia on flax straw are the major source of primary infection in France. Pasmo disease is commonly spread among flax cultivation areas over the world. It is known to reduce yield by 60 to 70% and to decrease the quality of seeds and fibers [1]. Some chemical options are available against pasmo in flax including QoI (quinone outside inhibitor), SDHI (succinate dehydrogenase inhibitor) and DMI-type (demethylation inhibitor) fungicides (Table 3A,B). To be efficient, these systemic products should be applied as early as possible during the infection, which is difficult to achieve since the symptoms sometimes occur only at the boll stage or even during ripening. The best option remains genetic resistance selected by breeding programs in flax. This is under investigation thanks to recent genetic prediction for QTL identification based on a GWAS (genome-wide association study) on 370 flax accessions from the Canadian core collection [60,61].

## 4. Soil-Borne Diseases

### 4.1. Fusarium wilt—Fusarium oxysporum f. sp. lini

*Fusarium oxysporum* f. sp. *lini* is a soil-borne fungus that infects flax plants via the roots. The fungus is host-specific and can impact flax cultivation at early stages, leading to the browning and delayed growth of seedlings or even to the senescence and death of the small flax plants. Infection can also occur in older plants, causing yellow/brown-colored spots on stems, leaves (Figure 4A) and buds, which subsequently become senescent and die. In an infected field, the impact of the disease is not homogeneous, which is characterized by the presence of brown spots within the field, as shown in Figure 4B. The apex of infected plants can turn downward, forming a crook. *F. oxysporum* colonizes flax xylem vessels, which leads to unilateral water deficiency symptoms that are visible on the side of infected vessels. Disease outbreaks can result in 80 to 100% yield losses, and the pathogen survives for decades in soil as chlamydospores. Isolates of *F. oxysporum* f. sp. *lini* can differ largely in aggressiveness. The origin of the pathogen is polyphyletic, and isolates from different parts of the world cluster in at least four distinct clonal lineages [62,63]. The most effective way to manage the disease is the use of resistant cultivars. Most modern flax varieties show moderate to high resistance to Fusarium wilt. Two QTLs associated with resistance to Fusarium wilt have been identified [64]. Recently, more insight was obtained into the mechanisms of resistance by a genome-wide association study, and 13 candidate genes involved in Fusarium wilt resistance were identified [17]. Some soils, such as the Châteaurenard soil in France, are naturally suppressive to Fusarium wilt. This calcic silt-clay soil contains 37.4% of CaCO_3_ and has a high pH (7.9). Suppressiveness is caused by the combined action of fluorescent *Pseudomonas* bacteria and non-pathogenic *F. oxysporum* that compete with pathogenic *F. oxysporum* for carbon and iron. Fusarium wilt of flax in a disease-conducive soil could be significantly reduced by the combined application of the non-pathogenic *F. oxysporum* strain Fo47 with the phenazine-producing fluorescent *Pseudomonas* strain [65]. *Bacillus subtilis* subsp. *spizizenii* strain ATCC 6633 also has a biocontrol potential against this fungus, and its efficiency to reduce Fusarium wilt of flax has been validated under controlled conditions [18]. European seed testing protocol [53] imposes the detection of *Fusarium* spp. in less than 5% of tested seeds to guarantee the certification of commercial flax seeds.

### 4.2. Sclerotinia Stem Rot—Sclerotinia sclerotiorum

This rot-causing pathogen is characterized by the water-soaking lesions (Figure 4E), bleaching and shredding of flax stems. The fungal mycelium grows on the surface of the infected stem inside which sclerotia (surviving form) are produced. *Sclerotinia sclerotiorum* is a problem for linseed in the UK [66] and for both flax types in Canada [67]. It has hundreds of host plants, and sclerotia can survive winter and adverse climatic conditions. Lodging increases the risk of infection, the soil-borne inoculum being more likely to infect fallen flax plants by contact. Therefore, for decades, the best way to avoid Sclerotinia stem rot has been to sow lodging-resistant cultivars [1]. This fungus is also involved in sunflower and rapeseed stem rot, species where genetic resistance has been recently found [68,69], opening the way for research on genetic resistance in flax. A biocontrol solution is the use of *Coniothyrium minitans*, a mycoparasite of *S. sclerotiorum* on various hosts, including lettuce, celery, sunflower, bean, oilseed rape and soybean [70,71]. Applied directly on the soil surface before seed sowing, it parasitizes sclerotia and impairs pathogen development by degrading fungal cells with cell-wall-degrading enzymes [72]. A biopesticide containing this mycoparasite is commercially available. Other biocontrol strategies to control *S. sclerotiorum* such as antibiosis, induced systemic resistance or hypovirulence mediated by mycoviruses have been intensively investigated and were reviewed by Albert et al. (2022) [73].

### 4.3. Scorch—Globisporangium megalacanthum and Berkeleyomyces basicola

Flax scorch is known to occur only in the coastal areas of Northern France, Belgium and the Netherlands [27]. This disease is characterized by the appearance of glossy lesions on brittle roots, which can lead to tissue necrosis and stunted growth of flax plants (Figure 4C). Leaves become brown and shriveled with senescence symptoms (Figure 4D). This disease is mainly due to the combinatorial effect of two pathogens, the oomycete *Globisporangium megalacanthum* (previously called *Pythium megalacanthum* [27]) and the ascomycete *Berkeleyomyces basicola* (previously called *Thielaviopsis basicola* or *Chalara elegans* [74]). *Globisporangium* spp. survive winter as resting structures called oospores. *B. basicola* produces chains of dark-colored chlamydospores and is known to cause black root rot on more than 170 agricultural and ornamental plant species [75]. A cold and wet climate favors the development of these pathogenic complexes, and late sowing is recommended to decrease the risk of flax scorch in fields. A few flax-scorch-resistant cultivars are commercialized since the early 2000s, and their use is highly recommended for cultivation within coastal areas. Biocontrol strategies using seed coating formulations with *Glomus intraradices,* an arbuscular mycorrhizal fungi (AMF), and antagonistic fungal strains (*Trichoderma atroviride*) could decrease scorch incidence in flax in greenhouse assays [76].

### 4.4. Seedling Blight/Root Rot—Rhizoctonia solani

Seedling blight is caused by a pathogen complex but predominantly by the basidiomycete *Rhizoctonia solani* Kühn (teleomorph: *Thanatephorus cucumeris* (A.B. Frank) Donk) [77]. This disease mainly occurs at the early stage of flax development, inducing typical red to brown lesions on the roots and hypocotyl (Figure 4H) just below the soil surface [78]. The seedlings attacked by *R. solani* often start to yellow, wilt and shrivel, and severe symptoms lead to the death of the flax plantlets [79]. The fungus might also attack flax plants after the flowering stage and induces root rot symptoms weakening these older plants [78]. Injuries of the roots make the plant considerably more susceptible to damage by root rot pathogens (such as *Pythium* and *Fusarium* species) and cold weather [79].

Current classification systems divide individual, multinucleate *R. solani* strains into 13 different anastomosis groups (AGs), based on hyphal fusion, culture morphology, rDNA-internal transcribed spacer sequences and pathogenicity [80]. Divergent studies revealed that AG 1, AG 2-1, AG 2-2, AG 4, AG 9 and to a lower degree AG 5 are the most aggressive anastomosis groups inducing seedling blight and/or root rot symptoms on flax [28,81,82,83,84,85,86,87]. Strains belonging to AG 3 only attack older plants, resulting in limited root rot [86], and AG 6 and AG 7 do not appear to be pathogenic [28]. The binucleate *Rhizoctonia* AG-E has also been isolated from older flax plants, which showed typical symptoms of root rot [85].

Due to the ability of *R. solani* to survive in the soil for a long time and cause disease in a broad range of plant species, sowing flax after alternate hosts is not recommended (e.g., sugar beets, leguminous crops, which are attacked by the same anastomosis groups as flax [88]). The disease can be controlled by using a combination of practices such as using high-quality seeds, incorporating a grass crop into the rotation with flax and sowing early in a well-prepared firm bed [79,89]. Brown-seeded linseed cultivars were found to be more tolerant to *R. solani* than yellow-seeded cultivars [90]. The application of fungicides is widely used for controlling *R. solani* on many crops, but the use of seed treatments to control flax seedling blight is not a common practice among flax growers [91], probably due to limited authorized fungicides.

### 4.5. Verticillium wilt—Verticillium dahliae

This vascular pathogen enters the plant through the roots and causes yellowing and senescence of leaves and stems, sometimes only on one half of the plant (Figure 4F) [1]. Symptoms appear from the bottom of the plant stem since the fungus is soil-borne. *Verticillium dahliae* can be present in soil for decades as long-lasting structures (microsclerotia). Infected flax plants can become senescent earlier than non-infected plants, and most of the time, the disease in the field does not produce any symptoms during the vegetative growth stage [30]. In addition to the strong persistence of the pathogen in soil, the wide range of host plants makes the primary inoculum pressure very difficult to decrease. In the field, the characteristic symptoms appear at the beginning of the retting process, when harvested plant stalks are spread in soil. At this stage, a gray/blue color appears in the stem of infected plants (Figure 4G). Microscopic black dots can also be visible, which are microsclerotia-producing spots [30]. The current increase in the frequency of this vascular disease leads to significant economic losses in flax cultivation since it particularly damages the fiber quality. Given the high survival ability of microsclerotia, the efficient removal of old flax residues, especially stalks and stubbles, and the cleaning of tools after each use are crucial points to avoid pathogen propagation within cultivated areas. Even if the genetic mechanisms sustaining the plant responses are studied and known, no genetic resistance against *V. dahliae* has been found so far in flax [92]. Biocontrol options against this fungus are studied with, for example, the biofumigation of soil in potato fields in Canada [93] or the use of antagonistic *Verticillium isaacii* strains, which have been tested in vitro and in fields to control *V. longisporum*, the causal agent of Verticillium wilt in cauliflower [94]. Efforts are also made to map and quantify *V. dahliae* populations in field soils, providing important information about the accurate location and the importance of fungal presence in fields [95].

## 5. Perspectives in Flax Disease Management

The best options to control soil- and seed-borne pathogens in flax are preventive measures aimed to exclude or eradicate the pathogen such as seed certification, cleaning and protection and cultural methods that reduce primary inoculum in soil. Strategies to protect flax plants from polycyclic pathogens include resistance breeding and foliar sprays with systemic fungicides.

### 5.1. Pathogen Characterization

Various flax diseases have been poorly studied, for some diseases, the causal agents are still unclear, and little is known about their genetic diversity and virulence mechanisms. Recently, DNA sequences that are frequently used for phylogenetic studies have been obtained for 203 flax fungal pathogens including various *Fusarium* spp. [17,96,97,98], *Colletotrichum lini* [99], *Aureobasidium pullulans* [99], *Septoria linicola* [23] and *Melampsora lini* [100,101]. These sequences can be used to develop PCR-based detection systems for flax pathogen identification [13] and are of great use for studying the virulence mechanisms of these pathogens and improving management strategies.

### 5.2. Sanitary Quality of Seeds

In 1931, the ISTA adopted the rules for seed testing (IRST) and, since 2012, flax seed production requires to follow these rules. To assess the level of contamination, in vitro incubation of flax seeds is performed in Petri dishes with nutritive media, according to Chapter 7 of the IRST protocol [4]. Three species of flax pathogens are targeted: *Alternaria linicola*, *Botrytis cinerea* and *Colletotrichum lini*. Flax seeds must be free of any of these three pathogens before commercialization. Since some soil-borne pathogens (*Globisporangium* sp., *Fusarium* spp., *Rhizoctonia* spp.) are likely to also infect the seed batches, an emergent practice is the cleaning of seeds before storing and sowing. Universal cleaning techniques are under development, such as physical treatment (hot air, hot water, electrons), as well as the use of biopesticides, to decrease the general level of pathogens that are carried by seeds. The ThermoSeed^®^ technology (Lantmännen, Sweden) is a commercial example of the hot pasteurization of seeds, which makes seeds free of any pathogen. It efficiently destroys *Fusarium* spp. in seeds from diverse crops like wheat, oat or barley, and since 2019, it is also successfully used in France to decontaminate flax seeds according to the ISTA rules. Authorized chemicals for seed coating in flax are limited (see Table 3B). Commercial biocontrol products, such as Integral^®^ Pro (BASF), use *Bacillus amyloliquefaciens* (MBI600 strain) in seed coating formulation as a general protection against fungal diseases of fiber flax and linseed.

### 5.3. Inoculum Reduction

In order to significantly decrease the inoculum of soil-borne flax pathogens in the field, a minimum period of seven years is required between two flax cultivations, during which no potential host for similar pathogens should be grown [102]. A strict cleaning of the farming tools between two different fields is also recommended, to avoid the presence of plant debris or soil which are sources of contamination by soil-borne pathogens, for example. For the same reason, plant residues must be removed from the field before starting any new flax cultivation. The development of diseases that are favored by wet environments, such as gray mold or pasmo, could also be prevented by controlling the number of weeds in the field [102]. In addition, the choice of the sowing period is a critical point since good climatic conditions ensure optimal germination and fast emergence of seedlings, reducing the risk of early disease development such as Alternaria or seedling blights. A prophylaxis approach is always recommended in flax fields, which requires early and precise detection of the presence of pathogens. However, early identification of flax soil-borne diseases can be difficult when symptoms are not visible before the late stages of plant development. For example, in the case of Verticillium wilt, symptoms appear after flowering or even only after harvest, during the retting stage, which is too late to counter the pathogen action.

Fumigation involves burying a chemical substance in the soil before seed sowing to effectively eliminate pathogenic microorganisms. In Europe, however, most chemical fumigants have been banned. Solarization can also be employed to decrease the soil inoculum using sunlight as a thermal treatment. For this, a black protective cover made of canvas is disposed on the soil, which increases the surface temperature up to 45 °C [103]. In some extreme cases, field soil is even excavated, heat-sterilized to kill pests and fungi and then put back into the field [104]. Interestingly, solarization has been successfully used to control *Verticillium dahliae* inoculum in olive tree orchards in southern Spain and, thus, constitutes an interesting alternative against this pathogen in flax culture [105]. Biofumigation is the same process as fumigation, but it involves the use of cruciferous crops that naturally contain high glucosinolate levels. Using them as green manure before sowing the crop of interest helps to control soil-borne pathogens present in a field, thanks to the volatile and toxic properties of the released molecules. Brown mustard is used, for example, to decrease the amount of *V. dahliae* in soil before potato cultivation [54]. However, these techniques are not pathogen-specific. A biopesticide based on *Coniothyrium minitans*, strain CON/M/91-08 (LALSTOP CONTANS ^WG^—Lallemand Plant Care), is commercially available in various countries to reduce *Sclerotinia* inoculum in soil.

### 5.4. Disease Suppressive Soils

The soil is divided into four areas depending on their distance from the plant root: the endosphere defines the area that is inside the root; the rhizoplane is the region where the root surface is in contact with soil, which is surrounded by the rhizosphere which is the soil area penetrated by plant roots, outside of which is the bulk soil area. The balance among the microbial communities from each soil area is very important for crop health [106]. When its microbiome is preserved, the soil can indeed become suppressive to diseases by the action of beneficial microorganisms, which promote antibiosis or competition toward plant pathogens [107]. The diversity of the microbiome also depends on other factors, including soil structure and content, temperature, nutrient status and pH, which are directly impacted by farming practices that play an important role in the emergence and maintenance of soil suppressiveness [108]. This is exemplified by the Fusarium-wilt-disease-suppressive soil from Châteaurenard that was discussed above [65]. In particular, the protection of the rhizosphere equilibrium by promoting bacterial beneficial communities turns out to be crucial. This has been shown in wheat cultivation to be the result of conservative agricultural practices (no-till farming), leading, for example, to the induction of suppressiveness to *Rhizoctonia* spp., a seed-borne pathogen impacting also flax fields [109]. The implementation of a strategy aiming at cultivating flax in suppressive soil would not be easy since it requires substantial efforts to develop specific farming methods. But it constitutes a very interesting perspective to achieve efficient and sustainable management of flax diseases in the near future, especially in the prospect of flax protection from soil-borne pathogens, such as *Fusarium oxysporum* and *Verticillium dahliae*, whose inoculum can survive for decades in soil. Various plant species have been reported to suppress Fusarium wilt disease in banana when used in rotation or in intercropping [110]. Recently, the incorporation of pineapple residues into the soil has been shown to stimulate antagonistic fungal populations, paving the way to new technical developments allowing improvement to the use of this natural microbe-based plant defense in agriculture. As structures of microbial soil communities are very complex, an alternative option is to use endophytes to control diseases without disturbing communities. This technique has been tested on several fungal diseases, such as banana’s Fusarium wilt (caused by *F. oxysporum* f. sp. *cubense*) [111] or turmeric’s rhizome rot and leaf blight, respectively caused by *Globisporangium aphanidermatum* and *Rhizoctonia solani* [112]. Endophytes such as *Verticillium isaacii* provide natural competition against *Verticillium longisporum* of cauliflower and also show great promise for flax cultivation protection against Verticillium wilt [113].

### 5.5. Chemical Control

Several chemicals are currently available to protect flax, for which details about active substances, the mode of action and diseases targeted are provided in Table 3A,B. Various QoI (quinone outside inhibitor), SDHI (succinate dehydrogenase inhibitor) and DMI-type (demethylation inhibitor) fungicides are authorized in flax, mainly to control powdery mildews, pasmo, Alternaria blight, Sclerotinia stem rot and gray mold. Foliar applications of fungicides are ineffective against soil-borne diseases like Fusarium and Verticillium wilts. Additionally, no active substance is efficient against scorch or Verticillium wilt, making the strategy of relying on chemical treatments for pathogens in flax cultivation an unsatisfactory approach. Moreover, the use of synthetic pesticides in plant disease management has generally decreased because of growing concerns on risks linked to residues in the environment. As described in a review by Palmieri et al. (2022) [114], the use of synthetic fungicides is also discouraged due to the following:An increasing amount of fungicide-resistant pathogen strains;A rising demand by consumers and plant product retailers for very low or even no synthetic chemical residues at all;More and more restrictive international regulations on the permitted levels of chemical residues in the environment and on the registration and eco-toxicological impact of pesticides (e.g., EU Directive 2009/128 on the sustainable use of pesticides and EU Green Deal 2019 Farm to Fork Strategy).Even if a global tendency tends toward a “zero” pesticide agriculture, another way is considered by researchers: integrated pest management. As described by Ons et al. (2020) [115] in a review, the increasing global food demand requires efficient ways to fight crop diseases. Combining synthetic pesticides and biocontrol products could be a suitable option.

### 5.6. Plant Breeding and Genetic Resistance

Plant breeding, using the natural genetic diversity for crop improvement, constitutes a powerful solution to produce pathogen-resistant cultivars, when a genetic resistance exists, such as the general pest and disease resistance of rice [116] or the qualitative resistance of tomato to *V. dahliae* [117]. In plants, genetic resistance to pathogens consists of either qualitative resistance, based on individual major resistance genes, or quantitative resistance, which is due to the simultaneous segregation of many genes involved in the reduction of pathogen multiplication. Qualitative resistance is related to effector-triggered immunity (ETI), and it is successfully used in directional selection programs to produce crops that are resistant to most pathogen strains. However, the emergence of a resistant pathogen strain irremediably leads to the collapse of such plant resistance and triggers severe epidemics. Quantitative disease resistance involves basal immunity or pathogen-associated molecular pattern (PAMP)-triggered immunity (PTI). It provides a lower level of immunity to the plant, causing the reduction but not the absence of disease (low symptoms can appear), and is driven by many QTLs, each one with small effects. This is a major type of disease resistance in many crop species, which presents the advantage of preventing pathogen bypass. Quantitative resistance is therefore more durable than gene-for-gene resistance but requires marker-assisted breeding programs to reach higher precision and effectiveness in targeting the related QTLs [118]. Most plant breeding programs are designed to benefit from the best combination of the two types of resistance, without compromising any other agronomic traits [116]. Flax displays a large genetic diversity reflected by several genetic collections. Among them, we find more than 11,000 accessions from 15 countries in the International Flax Database, more than 6000 accessions at the All-Russian Flax Research Institute and more than 5500 at the N. I. Vavilov Institute of Plant Industry stored in Russia; 3378 accessions from 76 countries are maintained at the Plant Genetic Resources of Canada. A reduced core collection is available that captures the diversity spectrum in the whole collection [6,48,119,120]. In flax, several qualitative and quantitative genetic resistances are systematically included in breeding programs, e.g., genetic resistance to rust in Canadian linseed cultivars and to scorch or powdery mildew and Fusarium wilt in European fiber flax. Genetic resistance to rust is mediated by several major genes (families K, L, M, N and P) through a gene-for-gene interaction between the pathogen (*avr* gene) and the host (*R* gene) [48,121]. To help face fungal diseases, flax cultivars are also selected for their ability to recover from lodging, since it increases the risk of infection with soil-borne pathogens such as *Sclerotinia* spp. Table 4 lists the location of resistance *loci* that have been identified in flax for some diseases cited in this review. Details on the gene families involved are available in Table A1 (Appendix A). Genome editing technologies, like CRISPR/Cas9 or base editing, combined with the availability of the flax genome sequence, could be implemented in genomic plant breeding to drastically increase the efficiency in the production of pathogen-resistant flax cultivars [122]. Current knowledge about genome editing tools and their applications to flax improvement has recently been reviewed by Clemis et al. in chapter 11 of “The Flax Genome” [48]. However, these technologies, like the use of genetically modified organisms, are not authorized in Europe and, thus, cannot be included in European flax breeding programs.

### 5.7. Induced Resistance

Another interesting perspective in the management of flax soil-borne diseases is the use of natural substances or microorganisms that induce plant protection against eukaryotic diseases. For example, the foliar application of leaf extract from the moringa tree enhances the production and efficiency of essential oils that are involved in marjoram (*Origanum majorana*) protection against *Fusarium* sp. [124]. Seaweed extract, with high plant defense elicitor content, is another example of a good candidate to improve plant defense efficacy against pathogens. Foliar application of such extracts induces an increase in wheat resistance against *Fusarium graminearum* [125]. Root drench of the pepper plant in an aqueous extract of sea bamboo showed a reduction in Verticillium wilt [125]. The use of plant-growth-promoting rhizobacteria (mainly *Bacillus* spp. and *Pseudomonas* spp.) is also studied to improve plant defense mechanisms and decrease disease incidence. These microorganisms produce molecules such as lactones, cyclic lipopeptides and rhamnolipids, which are used as plant defense elicitors in various pathosystems. Some examples are surfactins from *B. subtilis*, employed against *B. cinerea* in bean and tomato, or iturins from *B. amyloliquefaciens*, used against Verticillium wilt in cotton [126]. Foliar application of biocontrol products is also studied. For example, *Fusarium oxysporum* or *Botrytis cinerea* could be controlled in tomato by spraying natural extracts of arbuscular mycorrhizal fungi (AMF) [72]. AMF have been extensively studied during the last decade to check their potential for improving soil quality and, therefore, inducing plant defense against pathogens. AMF have shown protective abilities against wheat powdery mildew (*Blumeria graminis* f. sp. *tritici*) when applied just after seedling emergence in controlled conditions [73]. Other biocontrol options are studied through foliar application or seed coating: chitosan- and alginate-based formulations. These biopolymers showed capacities to increase plant defense mechanisms and decrease plant disease incidence [127,128].

## 6. Conclusions

Flax (*Linum usitatissimum*) is an important crop with a rich history of cultivation, but it is susceptible to various air-, soil- and seed-borne diseases that can cause significant yield losses. Preventive measures are good options to control soil- and seed-borne pathogens. Seed certification, cleaning and protection, along with cultural methods, contribute to the reduction in primary inoculum in soil. When feasible, solarization and biofumigation can also offer effective alternatives for reducing inoculum levels. The development of farming methods that promote soil suppressiveness seems to be a sustainable strategy, especially in the context of increasing flax-cultivating areas and decreasing the use of synthetic pesticides. Simultaneously, endophytes and other microorganism-based antagonists or plant resistance inducers show promising results in controlling flax eukaryotic diseases as well. Global climate change may modify plant disease dynamics, making research to understand plant–pathogen interactions a crucial task. On top of these strategies, breeding and sowing resistant cultivars constitute a good strategy to protect flax plants from polycyclic pathogens.

In conclusion, effective flax disease management requires a multifaceted approach that integrates preventive measures, genetic resistance and innovative biological solutions. Collaborative efforts between scientists, breeders and farmers will be essential in ensuring the future health and productivity of the flax crop.

## Figures and Tables

**Figure 1 plants-12-02811-f001:**
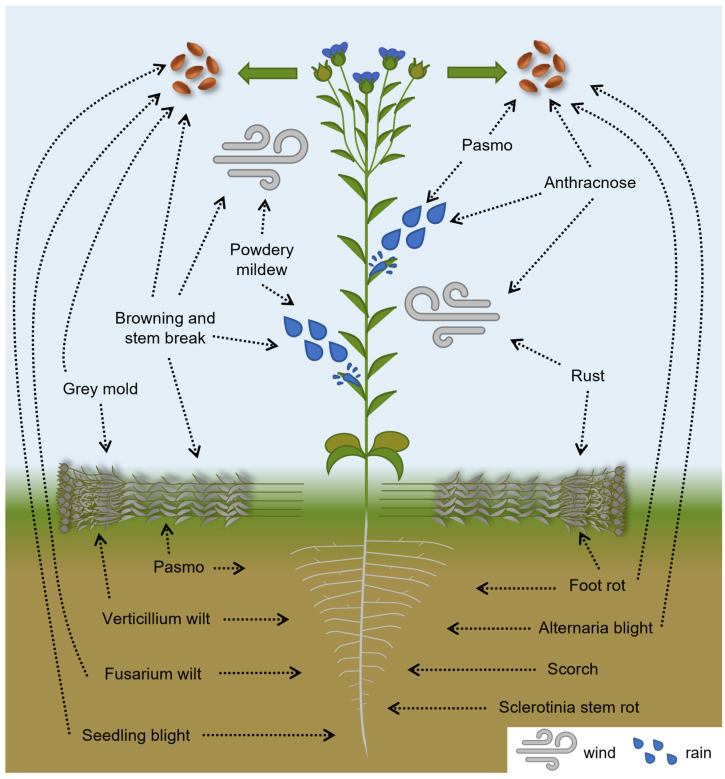
Flax plant diseases caused by eukaryotic pathogens and their transmission modes.

**Figure 2 plants-12-02811-f002:**
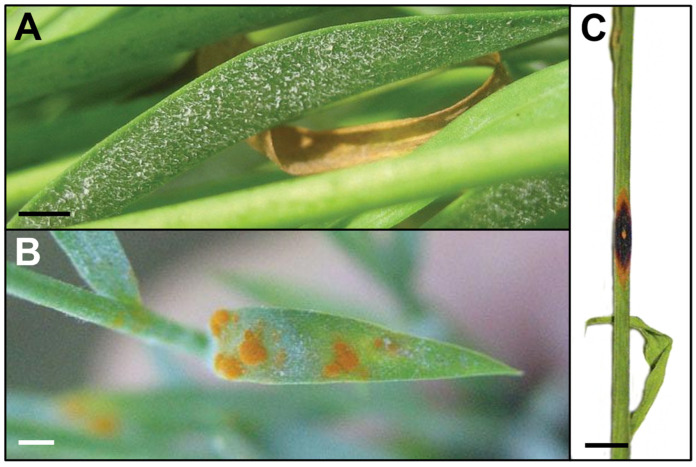
Symptoms of air-borne diseases in flax. (**A**) White powdery mildew (*Podosphaera lini*) colonies (mycelia) on flax leaf; bar = 0.2 cm. (**B**) Rust (*Melampsora lini*) orange powdery pustules (uredia) on flax leaf; bar = 0.2 cm. (**C**) Dark brown circular lesion of necrosis produced by rust (*Melampsora lini*) on flax stem; bar = 1 cm. (*Credits: Arvalis*).

**Figure 3 plants-12-02811-f003:**
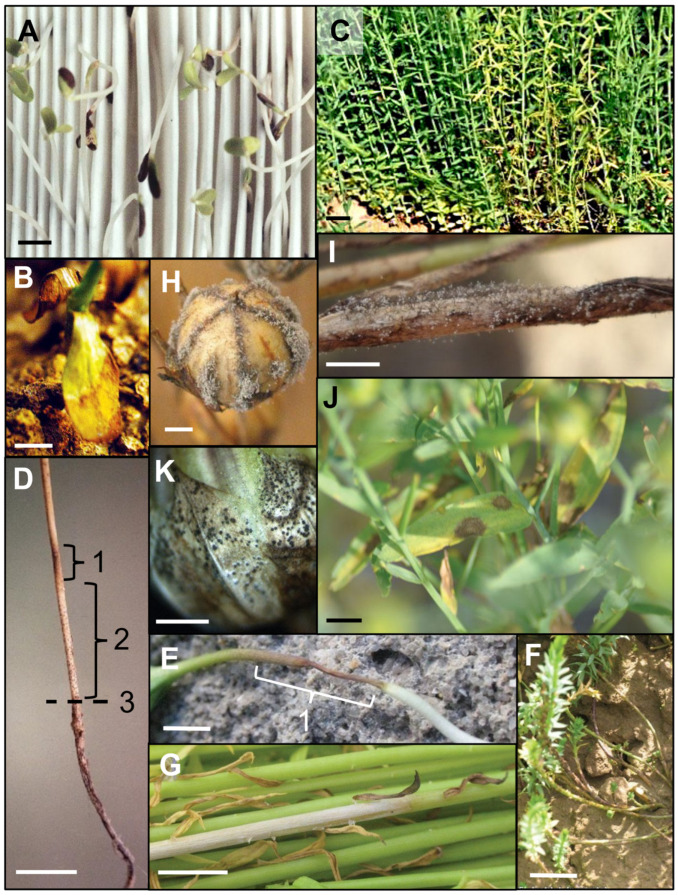
Symptoms of seed-borne diseases in flax. (**A**) Dark red spots of Alternaria blight (*Alternaria linicola*) on cotyledons and collar 3 days after germination of flax seedlings grown on wet paper; bar = 1 cm. (**B**) Anthracnose (*Colletotrichum lini*) symptoms in flax cotyledons; bar = 1 cm. (**C**) *Boeremia exigua* lesions on flax stems in the field, starting from a cluster of 5 to 10 plants; bar = 5 cm. (**D**) Details of *Boeremia exigua* symptoms in basal flax stem, above collar (3), displaying brown (1) and discolored (2) areas; bar = 1 cm. (**E**) Dark brown lesion (1) caused by *Aureobasidium lini* on field flax seedling (collet); bar = 0.5 cm. (**F**) Brown and broken field flax stems due to *Aureobasidium lini;* bar = 10 cm. (**G**) Gray color of flax stem due to the presence of *Botrytis cinerea* colonies; bar = 2 cm. (**H**) *Botrytis cinerea* gray mold on flax bolls; bar = 0.2 cm. (**I**) *Botrytis cinerea* gray, brown and black spots on decaying flax stem; bar = 0.5 cm. (**J**) Brown senescent areas in flax leaves due to *Septoria linicola;* bar = 0.5 cm. (**K**) Sepals of flax bolls with *Septoria linicola* black spots (pycnidia); bar = 0.2 cm. (*Credits: Arvalis*.)

**Figure 4 plants-12-02811-f004:**
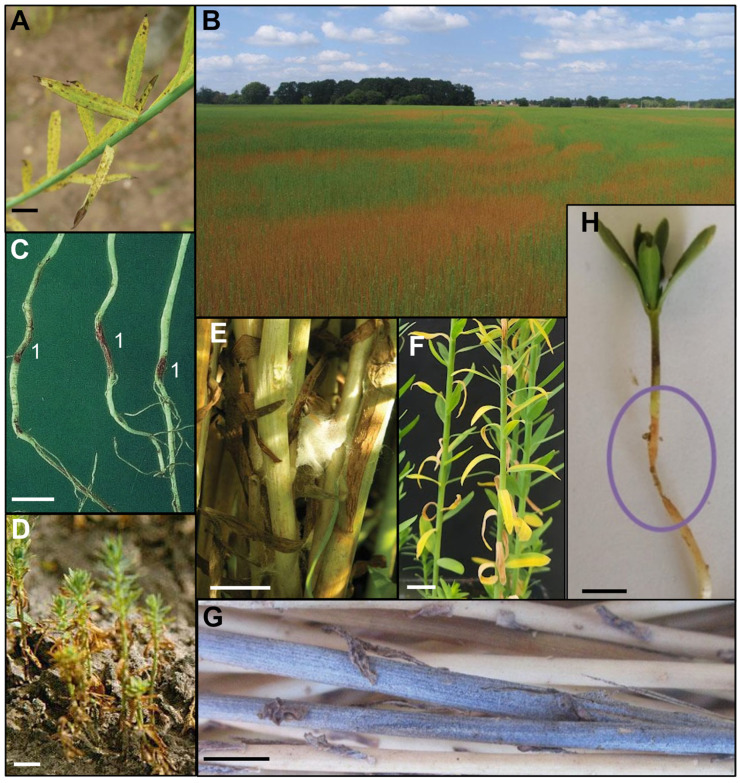
Symptoms of soil-borne diseases in flax. (**A**) Field flax leaves with yellowing and brown spots resulting from the vascular infection by *Fusarium oxysporum* f. sp. *lini*; bar = 0.5 cm. (**B**) Flax field affected by Fusarium wilt, as shown by brown areas where plants are infected. (**C**) In vitro roots from 20-day-old flax plants displaying dark brown scorch (*Globisporangium megalacanthum* and *Berkeleyomyces basicola*) (1) lesions; bar = 1 cm. (**D**) Twenty-day-old flax plantlets with stunted growth caused by scorch; bar = 2 cm. (**E**) Field flax stems and leaves infected by *Sclerotinia sclerotiorum* and showing water-soaked lesions and bleached color; bar = 0.5 cm. (**F**) Greenhouse flax plants (30 days old) with half yellowing and senescent leaves, disposed along a gradient from bottom to the top, resulting from Verticillium wilt infection; bar = 1 cm. (**G**) Retting flax stems showing blue/gray color and *Verticillium dahliae* microsclerotia (microscopic black spots); bar = 0.5 cm. (**H**) *Rhizoctonia solani* causing shriveling and wilting (circle) of 10-day-old flax plantlet; bar = 1 cm. (*Credits: A–E*, *G*, *Arvalis*; *F*, *Julie Moyse*).

**Table 1 plants-12-02811-t001:** **A.** Total production, acreage and yield of linseed (*Linum usitatissimum* L.) in the top ten producing countries in 2021 [3]. **B.** Total production, acreage and yield of fiber flax (*Linum usitatissimum* L.) in the top ten producing countries in 2021 [3].

Ranking	Country or Area	Total Production (ton)	Acreage (ha)	Crop Yield (ton/ha)
**A**				
1	Russia	1,300,232	1,492,119	0.9
2	Kazakhstan	775,517	1,366,068	0.6
3	Canada	345,719	403,500	0.9
4	China, mainland	340,002	260,000	1.3
5	India	111,007	183,725	0.6
6	Ethiopia	82,000	80,000	1.0
7	France	72,941	37,500	1.9
8	United Kingdom and Northern Ireland	70,998	40,907	1.7
9	United States of America	68,785	108,460	0.6
10	Ukraine	42,231	27,600	1.5
**B**				
1	France	678,385	112,580	6.0
2	Belgium	87,000	15,390	5.7
3	Belarus	35,680	42,300	0.8
4	China, mainland	27,130	6317	4.3
5	Russia	25,947	36,483	0.7
6	United Kingdom and Northern Ireland	14,745	10,095	1.5
7	The Netherlands	11,330	1800	6.3
8	Egypt	7601	8609	0.9
9	Chile	3065	2769	1.1
10	Argentina	2620	2904	0.9

**Table 2 plants-12-02811-t002:** Causal agents of major eukaryotic diseases in flax and affected organs.

Disease Common Name	Causal Agent		NCBI Taxonomy ID EPPO Code	Lifestyle	Affected Organs *	References
	Phylum	Species				
Anthracnose	Ascomycota	*Colletotrichum lini* (*Colletotrichum linicola*)	500171 COLLLI	Hemibiotrophic	Seedling Leaf Stem Buds/Flowers	[1,10,11,12,13]
Alternaria blight	Ascomycota	*Alternaria linicola*	75319 ALTELI	Necrotrophic	Seedling Stem Bolls/Seeds	[1,14]
Browning and stem break	Ascomycota	*Aureobasidium pullulans* var. *lini* (*Kabatiella lini, Polyspora lini, Discosphaerina fulvida, Guignardia fulvida*)	1836197 AUREPL	Necrotrophic	Seedling Stem	[1,13,15]
Foot rot/basal stem blight	Ascomycota	*Boeremia linicola* (*Phoma exigua* f. sp. *linicola*)	2904145 PHOMEL	Necrotrophic	Root Seedling Stem Bolls/Seeds	[1,16]
Fusarium wilt	Ascomycota	*Fusarium oxysporum* f. sp. *lini*	120040 FUSALI	Hemibiotrophic	Seedling Leaf Stem Buds/Flowers	[1,11,17,18]
Gray mold	Ascomycota	*Botrytis cinerea*	40559 BOTRCI	Necrotrophic	Seedling Bolls/Seeds	[1,11,19,20]
Pasmo	Ascomycota	*Septoria linicola* (*Mycosphaerella linicola*)	215465 MYCOLN	Hemibiotrophic	Seedling Leaf Stem Bolls/Seeds	[1,21,22,23,24]
Powdery mildew	Ascomycota	*Podosphaera lini* (*Oidium lini*), *Golovinomyces orontii* and others	683379 62715 ERYSPP	Biotrophic	Leaf Stem	[1,25]
Rust	Basidiomycota	*Melampsora lini*	5261 MELMLI	Biotrophic	Leaf Stem Bolls/Seeds	[1,11]
Sclerotinia stem rot	Ascomycota	*Sclerotinia sclerotiorum*	5180 SCLESC	Necrotrophic	Stem	[1,26]
Scorch	Oomycota Ascomycota	*Globisporangium megalacanthum* (*Pythium megalacanthum*) and *Berkeleyomyces basicola* (*Thielaviopsis basicola, Chalara elegans*)	147705 PYTHME 124036 THIEBA	Necrotrophic Hemibiotrophic	Root Seedling Leaf Stem	[1,27]
Seedling blight/root rot	Basidiomycota	*Rhizoctonia solani*	456999 RHIZSO	Necrotrophic	Root Seedling Stem	[1,28,29]
Verticillium wilt	Ascomycota	*Verticillium dahliae*	27337 VERTDA	Hemibiotrophic	Leaf Stem Windrow	[1,12,30]

* Symptoms will be described and illustrated further in this review.

**Table 3 plants-12-02811-t003:** **A.** Fungicide active substances that are authorized for flax leaf treatment **B.** Fungicide active substances that are authorized for flax seed treatment in the most important flax-producing countries against pathogens mentioned in this review (available data only, [39,40,41,42,43,44,45]).

FRAC Acknowledged Mode of Action— Target Site Code	Chemical Group— FRAC Code	Active Substance	Resistance Risk	Target Disease *	Areas of Authorization
**A**					
Contact fungicide action					
Chemicals with multi-site activity	Inorganic—M02	Sulfur	Low	Powdery mildew	BE
Data not available	Inorganic	Potassium hydrogen carbonate	Data not available	Powdery mildew	NL
Signal transduction—E2	Phenylpyrroles—12	Fludioxonil ^a^	Low to medium	Gray mold	NL
Unknown	Phenyl-acetamides—U06	Cyflufenamid	Unknown	Powdery mildew	BE, FR
Systemic fungicide action					
Cytoskeleton and motor protein—B1	MBC—Thiophanate—1	Thiophanate-methyl	High	Anthracnose Fusarium wilt Pasmo	BY
Respiration—C2	SDHI—pyrazole-4-carboxamides—7	Benzovindiflupyr ^b^	Medium to high	Pasmo Powdery mildew	GB, IE
		Bixafen		Sclerotinia stem rot	NL, CA
		Fluxapyroxad ^c^			
	SDHI—pyridine-carboxamides—7	Boscalid	Medium to high	Basal stem blight Foot rot	FR
				Powdery mildew	NL
	SDHI—pyridinyl-ethyl-benzamides—7	Fluopyram ^d^	Medium to high	Gray mold Sclerotinia stem rot	NL
Respiration—C3	QoI—methoxy-acrylates—11	Azoxystrobin ^e^	High	Anthracnose	BY
				Browning and stem break	BY, FR
				Fusarium wilt	BY
				Pasmo	BY, CA, FR
				Powdery mildew	CA, FR
				Sclerotinia stem rot	CA, FR, NL
		Picoxystrobin	High	Pasmo	CA
	QoI- methoxy-carbamates—11	Pyraclostrobin ^c^	High	Anthracnose	BY
				Fusarium wilt	BY
				Pasmo	BY, CA
				Powdery mildew	NL
				Sclerotinia stem rot	CA
	QoI—oximino-acetates—11	Trifloxystrobin	High	Alternaria blight Powdery mildew	NL
Amino acid and protein synthesis—D1	AP—anilino-pyrimidines—9	Cyprodinil ^a^	Medium	Gray mold	NL
Sterol biosynthesis in membranes—G1	DMI—Triazoles—3	Cyproconazole	Medium	Anthracnose Fusarium wilt Pasmo	BY
		Difenoconazole ^e^	Medium	Anthracnose	BY
				Basal stem blight	FR
				Browning and stem break	FR
				Foot rot	FR
				Fusarium wilt	BY
				Pasmo	BY, CA, FR
				Powdery mildew	BE, CA, FR, NL
				Sclerotinia stem rot	CA, FR
		Epoxiconazole	Medium	Anthracnose Fusarium wilt Pasmo	BY
Systemic fungicide action					
Sterol biosynthesis in membranes—G1	DMI—Triazoles—3	Flutriafol	Medium	Anthracnose Fusarium wilt Pasmo	BY
		Propiconazole	Medium	Anthracnose Browning and stem break Pasmo	BY
		Spiroxamine	Low to medium	Anthracnose Fusarium wilt	BY
		Tebuconazole ^d^	Medium	Anthracnose	BY
				Browning and stem break	FR
				Fusarium wilt	BY
				Gray mold	GB, IE, NL
				Pasmo	FR
				Powdery mildew	BE, FR
				Sclerotinia stem rot	NL
	DMI—Triazolinthione—3	Prothioconazole ^b^	Medium	Anthracnose	BY
				Fusarium wilt	BY
				Pasmo	GB, IE
				Powdery mildew	BE, FR, GB, IE
				Sclerotinia stem rot	CA
**B**					
Contact fungicide action					
signal transduction—E2	Phenylpyrroles—12	Fludioxonil	Low to medium	Alternaria blight	BE, NL
				Anthracnose	NL
				Basal stem blight	BE, NL
				Foot rot	BE, NL
				Fusarium wilt	BE, FR, NL
				Gray mold	BE
				Sclerotinia stem rot	BE, NL
				Seedling blight	NL
chemicals with multi-site activity	Dithiocarbamates—M03	Thiram ^f^	Low	Fusarium wilt Seedling blight	CA
Systemic fungicide action					
nucleic acid metabolism—A1	PA—Acylalanines—4	Metalaxyl-M	High	Alternaria blight	NL
				Anthracnose	NL
				Basal stem blight	NL
				Foot rot	NL
				Sclerotinia stem rot	NL
				Scorch	BE, FR
				Seedling blight	NL
cytoskeleton and motor protein—B1	MBC—Benzimidazoles—1	Carbendazim	High	Anthracnose Fusarium wilt Pasmo	BY
respiration—C2	SDHI—Oxathiin-carboxamides—7	Carboxin ^a^	Medium to high	Fusarium wilt Seedling blight	CA

* For details on causal agents, see also Table 2. ^a^ used in combination with cyprodinil/fludioxonil; ^b^ used in combination with prothioconazole/benzovindiflupyr; ^c^ used in combination with pyraclostrobin/fluxapyroxad; ^d^ used in combination with tebuconazole/fluopyram; ^e^ used in combination with difenoconazole/azoxystrobin; ^f^ used in combination with carboxin BE: Belgium; BY: Belarus; CA: Canada; FR: France; GB: Great Britain; IE: Ireland; NL: The Netherlands; MBC: Methyl Benzimidazole Carbamate; SDHI: Succinate Dehydrogenase Inhibitor; QoI: Quinone Outside Inhibitor; AP: Anilino-Pyrimidine; DMI: Demethylation Inhibitor; PA: Phenylamides.

**Table 4 plants-12-02811-t004:** List of most recent published genetic resistances in plants against eukaryotic flax pathogens. Details on gene families involved are available in Appendix A Table A1.

Causal Agent (Strain)	Cultivar/Type	Plant Cultivation Details	Disease Assessment Details	Technique—Results	Chr Involved *	Reference
Alternaria blight *Alternaria linicola*	F_2_ population (disease-resistant genotype × disease-susceptible genotype)	Field (India)—2014	Disease rating Score: 0–5 Stages: flowering harvesting	Simple sequence repeats (SSR) genotyping—2 QTLs	14	[51]
Fusarium wilt *Fusarium oxysporum* f.sp. *lini* (*MI39*)	Russian flax genetic collection (179 fiber flax accessions and 117 linseed accessions)	Greenhouse—3 years in a row (2019–2021)—Infection of soil using pure culture of fungus	Disease Severity Index Score: 0–3 Stage: early yellow ripeness	GWAS— 15 QTNs	1, 8, 11, 13	[17,48]
Powdery mildew *Oïdium lini*	F_3_ and F_4_ populations (disease-resistant genotype × disease-susceptible genotype)	F_3_: field (Canada, 2012), under natural powdery mildew infection F_4_: growth chamber, infection with isolate PM97 propagated on flax plants in controlled conditions	Disease rating Score: 0–9 Stages: F_3_: end of flowering/mid- and late green boll F_4_: pre-flowering	Simple sequence repeats (SSR) genotyping—3 QTLs	1, 7, 9	[123]
	Worldwide collection (38 countries) 173 fiber flax 110 linseed 2 wild flax 19 unknown type flax	Field (France)—2018 and 2019 Natural field infection	Disease rating with AUDPC Score: 0–9 Stage: flowering	GWAS—10 QTLs	1, 2, 4, 13, 14	[38]
	Worldwide collection (39 countries) 80 fiber flax 292 linseed	Field (Canada)—2012 to 2016 Greenhouse-inoculated diseased plants (10) every ten rows	Disease rating Score: 0–9 Stages: early flowering (8 to 9 weeks after sowing) late flowering (9 to 10 weeks after sowing) green boll (10 to 11 weeks after sowing) early brown boll (11 to 12 weeks after sowing)	GWAS— 349 QTNs	All	[37,48]
Pasmo *Septoria linicola*		Field (Canada)—2012 to 2016 Spread of infested chopped straw between rows and misting		GWAS— 692 QTNs	All	[48,58,60]

* Details on gene families involved (when data are available) are presented in Appendix A Table A1. Chr: chromosome; AUDPC: area under disease progress curve; GWAS: genome-wide associated studies.

## Data Availability

Not applicable.

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
