# Peer review of "Overview and Management of the Most Common Eukaryotic Diseases of Flax (Linum usitatissimum)"

_plants, 2023, doi:10.3390/plants12152811_

Round 1

Reviewer 1 Report

This is a very well-structured and informative introduction to the paper. It clearly outlines the importance of flax, summarizes and summarizes the many fungal diseases of this cash crop, details the characteristics of the different diseases and the means of control, and lays the foundation for the disease management of flax.

Here are a few minor suggestions that could potentially improve the text:

  1. Consider removing redundant phrases or details. For example, "annually cultivated crop" can be reduced to "annual crop."
  2. Ensure that all the numbers provided (for example, regarding production levels or areas) are correctly attributed and referenced. If these figures result from published research, state this explicitly.
  3. Consider mentioning earlier in the introduction that the review will be focused on eukaryotic diseases that affect flax. This will help orient the reader right from the start.
  4. Please evaluate the clarity of the text. To make it more straightforward, consider rephrasing sentences such as "Flax bolls are harvested after approximately 120 days of cultivation in the field, by a combine harvester" to "After around 120 days of cultivation in the field, a combine harvester is used to harvest flax bolls."
  5. Try to ensure that the writing is clear and concise. For instance, "Emergence of this disease is favored by hot and dry conditions occurring during the end of spring, when flax is in the pre-flowering stage" could be simplified to "This disease emerges under hot and dry conditions, particularly towards the end of spring when flax is in the pre-flowering stage."
  6. Several complex and technical terms in sections might need to be explained to a non-specialist reader. If your audience includes such readers, consider including brief definitions or explanations of these terms. 
  7. Providing more specific information about the resistance mechanisms against these diseases in flax might be helpful. You've done an excellent job of this in the rust section, but it could also be helpful in the powdery mildew section. For instance, you could explain more about how the Pm1 gene and the other identified genes provide resistance against powdery mildew.
  8. When describing pathogenic fungi, additional information on morphological characteristics and classification of pathogenic fungi, such as morphological diagrams, could be provided, which may help the reader identify them. 
  9. When discussing control measures, more specific information can be described. For example, a detailed description of disease-resistant varieties' screening and selection process and the mechanisms of action of QTL and candidate genes associated with Fusarium wilt resistance have been identified.

Please make your own changes to the language descriptions in your article so that the reader can better understand the point you are trying to make.

Author Response

Dear reviewer,

Please see the attachment. The quality of the figures in this file is not good, due to impairment during the fusion process of two pdf files (comments and revised ms) which was required to make the single attached pdf file. The revised manuscript will include top quality figures.

Best regards,

Laurent Gutierrez

Reviewer 2 Report

The manuscript: “Overview and management of the most common eukaryotic diseases on flax (Linum usitatissimum)” consists of the interesting and comprehensive description of diseases dangerous for flax. Moreover, it also indicates chemical and non-chemical methods of reducing these diseases. The manuscript is well written, contains tables and very good photographs which are well described. The only my suggestion is that the authors should add the chapter Conclusions, where they could write a few sentences about future perspectives for management of the diseases in flax. What are the predictions of the authors. Will the described diseases pose less/greater risk to flax? What are the prospects for further disease reduction (for example: more resistant varieties?) What are the needs/expectations for flax cultivation?

Author Response

Dear reviewer,

Please see the attachment. The quality of the figures in this file is not good, due to impairment during the fusion process of two pdf files (comments and revised ms) required to make the single attached pdf file. The revised manuscript will include top quality figures.

Best regards,

Laurent Gutierrez
